# Optimal Physical Activity Is Associated with the Reduction of Depressive Symptoms via Neuroticism and Resilience

**DOI:** 10.3390/healthcare11131900

**Published:** 2023-06-30

**Authors:** Kazuki Nakajima, Akiyoshi Shimura, Masayuki Kikkawa, Shunichiro Ito, Mina Honyashiki, Yu Tamada, Shinji Higashi, Masahiko Ichiki, Takeshi Inoue, Jiro Masuya

**Affiliations:** 1Department of Psychiatry, Tokyo Medical University, 6-7-1 Nishi-Shinjuku, Shinjuku-ku, Tokyo 160-0023, Japan; 2Department of Psychiatry, Gakuji-Kai Kimura Hospital, 6-19 Higashi-Honcho, Chuo-ku, Chiba 260-0004, Japan; 3Department of Psychiatry, Tokyo Medical University Hachioji Medical Center, 1163 Tatemachi, Hachioji-shi, Tokyo 193-0998, Japan; 4Department of Psychiatry, Tokyo Medical University Ibaraki Medical Center, 3-20-1 Chuo, Ami-machi, Ibaraki 300-0395, Japan

**Keywords:** physical activity, exercise, resilience, neuroticism, depressive symptoms, path analysis

## Abstract

Background: Personality traits, such as neuroticism, that results in vulnerability to stress, and resilience, a measure of stress coping, are closely associated with the onset of depressive symptoms, whereas regular physical activity habits have been shown to reduce depressive symptoms. In this study, the mediating effects of neuroticism and resilience between physical activity duration and depressive symptoms were investigated by a covariance structure analysis. Methods: Between April 2017 and April 2018, 526 adult volunteers were surveyed using self-administered questionnaires. Demographic information, habitual physical activity duration (PAD), neuroticism, and resilience were investigated. The effects of these factors on depressive symptoms were analyzed by a covariance structure analysis. This study was conducted with the approval of the Medical Ethics Committee of Tokyo Medical University. Results: The dose–response curves of physical activity duration and depression scores were U-shaped: the optimal physical activity duration for the lowest depression score was 25.7 h/week. We found that the greater the difference from the optimal PAD, the higher the neuroticism and the lower the resilience, and the more severe the depressive symptoms. Covariance structure analysis demonstrated that neuroticism and resilience significantly and completely mediated the effects of the difference from the optimal PAD on depressive symptoms (coefficient of determination *R*^2^ = 0.349). Conclusion: Our study suggests that there is an optimal PAD that reduces depressive symptoms, and that a greater difference from the optimal PAD increases depressive symptoms through neuroticism and resilience.

## 1. Introduction

Previous studies have shown that higher physical activity is associated with lower levels of anxiety and depressive symptoms [1]. Exercise tailored to an individual’s ability can reduce the severity of depression [2]. Meta-analyses have demonstrated the efficacy of exercise interventions in treating depression among patients [3,4]. Furthermore, physical activity is reportedly associated with a reduced risk of developing depression [5,6], and even low-intensity physical activity may prevent depression [7]. As these studies have shown, the association of exercise/physical activity and the prevention and treatment of depressive symptoms and depression is becoming more clear, but the underlying mechanisms by which exercise/physical activity improves depressive symptoms and depression have not been fully identified to date.

Resilience is defined as “the process of adapting in the face of adversity, trauma, tragedy, threat, or even significant stress” [8]. Prior research has shown that resilience is a crucial therapeutic target for treating psychiatric disorders, such as anxiety and depression [9]. Resilience is negatively associated with depressive symptoms [10]. Intervention studies have also found that promoting resilience can reduce stress-induced depressive symptoms [11], and that cognitive behavioral therapy aimed at increasing resilience can improve symptoms of depression [12]. Research by Kermott and colleagues has also suggested that low resilience is a risk factor for the development of depression [13], and that resilience may impact the severity and onset of depressive symptoms.

Neuroticism is a personality trait with a tendency to negatively respond to stimuli, such as anger, anxiety, and depression [14], and is an important risk factor for the onset of depression [15]. Recent studies have reported that neuroticism is a risk factor for the onset of depression [16,17], and plays a mediating role in the association between adverse childhood experiences and depressive symptoms in the general adult population [18,19]. Thus, it has been suggested that neuroticism is an important individual risk factor for the treatment and prevention of depression.

Physical activity, resilience, and neuroticism also affect each other. Persons with positive attitudes toward exercise have low neuroticism [20], and regular exercise leads to low neuroticism [21]. Regular moderate-to-vigorous physical activity is associated with lower neuroticism, anxiety, and depressive symptoms [21,22]. Furthermore, physical activity is positively associated with resilience [23,24,25], and relatively rigorous exercise indirectly affects depressive symptoms through its effects on social support and resilience [26]. Physical activity is suggested to facilitate resilience through strengthening both individual brain regions as well as large-scale neural circuits to improve emotional and behavioral regulation [27]. On the other hand, there have been no reports on how resilience and neuroticism play roles in the positive effect of physical activity on depression. Recently, our research group found that optimal physical activity lowered depressive symptoms in adult volunteers, and this effect was mediated by the reduction of trait anxiety, which is a personal trait susceptible to depression [28]. Similar to trait anxiety, neuroticism and resilience directly influence depressive symptoms, and physical activity can improve neuroticism and resilience. Changes in neuroticism and resilience might mediate the antidepressant effect of physical activity. In addition, although a recent study indicated that an optimal physical activity duration exists regarding the impact of physical activity intensity on mental health [29,30], the association of an optimal physical activity duration with resilience, neuroticism, and depressive symptoms has also not been investigated to date.

Therefore, we focused on the mediating effects of neuroticism and resilience and hypothesized that there is an optimal habitual physical activity duration regarding its effect on depressive symptoms, and that the effects of habitual physical activity on depressive symptoms are mediated by its impact on neuroticism and resilience. To test this hypothesis, we conducted a cross-sectional study using a questionnaire survey on adult volunteers and tested the hypothesis via a path analysis. The purpose of this study was to test the mediating effects of neuroticism and resilience between optimal habitual physical activity and the severity of depressive symptoms in adult volunteers.

## 2. Subjects and Methods

### 2.1. Subjects

A self-administered questionnaire was distributed to adult volunteers between April 2017 and April 2018. Written consent was obtained from the subjects, and demographic information and three questionnaires were anonymously surveyed. Data were collected by a paper-based questionnaire survey without interviews, and questionnaires were returned anonymously by mail. Subjects were recruited by convenience sampling through our acquaintances, who were workers and their relatives, at Tokyo Medical University. Additionally, we aimed to collect a sample size of 651 individuals in order to detect an effect size of d = 0.2 with a 95% significance level. Valid responses were obtained from 526 subjects (228 men and 298 women; mean age: 41.2 ± 11.9 years; age range 20 to 77). The study participants were informed that their participation was completely voluntary, and that they would not experience any negative consequences for choosing not to participate. They were also reassured that all collected data would be kept anonymous, ensuring that individuals could not be personally identified. The inclusion criterion was being an adult. The exclusion criterion was having a severe physical or psychiatric disease. Subjects were generally healthy and had no severe diseases influencing their life functions. Therefore, this exclusion criterion excluded no subjects from the recruited subjects. This study was conducted with the approval of the Medical Ethics Review Committee of Tokyo Medical University (study approval No. SH3502). This dataset is similar to the previously published studies [28,30].

### 2.2. Questionnaires

Age, sex, education years, current marital status (yes/no), current employment status (yes/no), subjective social status (1 to 10) [31], existence of a past history of psychiatric or physical disease (yes/no), and first-degree relative with a psychiatric disease (yes/no) were assessed as demographic variables.

The shortened version of the neuroticism subscale, the Eysenck Personality Questionnaire-revised (EPQ-R) [32], was used to assess neuroticism. This subscale consists of 12 items, and each item is assessed using a 2-point scale of “1: yes” and “0: no”. The total score (range: 0–12) of the questionnaire was used for the study analyses. Our previous research has established the reliability and validity of the Japanese version of the EPQ-R [33].

The Connor-Davidson Resilience Scale (CD-RISC) [9] has been widely used in recent years to assess resilience, and the Japanese version of the CD-RISC was used in this study. Its reliability and validity have been confirmed in previous research [34]. The CD-RISC comprises 25 items, which are rated on a 5-point scale (0: not true at all, 1: rarely true, 2: sometimes true, 3: often true, and 4: true nearly all of the time) based on how the subject has felt over the past month, and the total score (range: 0–100) was used for the analysis in this study.

The Patient Health Questionnaire-9 (PHQ-9) is a self-administered depression rating scale [35]. Depressive symptoms in the previous 2 weeks are assessed. The PHQ-9 consists of 9 items, and each item is assessed using a 4-point scale (0: not at all, 1: several days, 2: more than half the days, and 3: nearly every day). In this study, the total score (range: 0–27) of the Japanese version of the PHQ-9 was used for analysis. The translation and validation of the Japanese version of the PHQ-9 for reliability and validity were conducted by Muramatsu et al. [36].

### 2.3. Physical Activity

The International Physical Activity Questionnaire (IPAQ) is a self-administered questionnaire widely used around the world [37]. The validated Japanese short form was used in this study [38]. The short form of IPAQ assesses the following three types of habitual physical activities: walking (in daily life and at work), moderate-intensity physical activity (carrying light loads, slow swimming, etc.), and vigorous-intensity physical activity (carrying heavy loads, jogging, etc.). The IPAQ enables the estimation of weekly total physical activity intensity (METs × minutes/week: METs = metabolic equivalents [39]), and weekly total physical activity duration (minutes/week). In this study, the habitual weekly total physical activity duration (PAD), which is the sum of the physical activity duration of each intensity, was used for the analysis.

### 2.4. Statistical Analysis

To analyze the association between the demographic information and the questionnaire data, Pearson’s correlation coefficient analysis, the *t*-test, and multiple regression analysis were conducted using SPSS Statistics Version 28 software (IBM, Armonk, NY, USA).

For PAD, a quadratic equation model introducing a squared term was also calculated, in addition to simple linear regression analysis, as it was assumed that there was an optimal time that minimized depression, rather than a linear response to depression.

Mplus version 8.5 software (Muthén and Muthén, Los Angeles, CA, USA) and robust maximum likelihood estimation with covariance structure analysis were used for the path analysis. All coefficients of the covariance structure analysis were standardized. Goodness-of-fit indices were not used because the model was a saturated model. A *p*-value of less than 0.05 was considered to indicate statistical significance. This study focused on the mediating effects of neuroticism and resilience and hypothesized an indirect effect of the difference from the optimal PAD (DOP, hours/week) on depressive symptoms through neuroticism and resilience. Based on this hypothesis, a path model was developed.

## 3. Results

### 3.1. Measured Variables

Table 1 shows the results of the analysis of the association between the demographic information and the data from each questionnaire and the PHQ-9 in 526 adult subjects. The mean PAD was 10.4 h/week (SD = 14.3 h/week). The value of Cronbach’s α of the PHQ-9 was 0.854, that of the CD-RISC was 0.948, and that of the EPQ-R was 0.859, all indicating a high reliability [40].

### 3.2. Association of PHQ-9 Scores with Demographic and Clinical Data and Questionnaire Data of the Study Population

The association between the PAD and the PHQ-9 scores was insignificant in linear regression (PHQ-9 = −0.000244 PAD + 3.974, *F* = 0.000, *p* = 0.985), whereas it was significant in the quadratic model (PHQ-9 = 0.002334 PAD^2^ − 0.119965 PAD + 4.492 = 0.002334 (PAD − 25.7)^2^ + 2.950, *F* = 6.906, *p* < 0.001). Table 2 shows the results of this multiple regression analysis. In Figure 1, the horizontal axis represents the PAD, and the vertical axis represents the PHQ-9 scores. From this quadratic equation, the PAD with the lowest PHQ-9 score, i.e., the lowest depressive symptoms, was 25.7 h per week, which was considered the optimal PAD. Following a previous study [30], we defined the difference in the PAD from the optimal time as the variable DOP.

Significant associations with high PHQ-9 scores were found for women, being unmarried, short education years, low subjective social status, past history of psychiatric disease, current psychiatric disease, EPQ-R neuroticism, CD-RISC, and DOP. On the other hand, there were no significant associations with PHQ-9 scores for age, employment status, current physical disease, and having a first-degree relative with a psychiatric disease.

### 3.3. Multiple Regression Analysis with PHQ-9 as the Dependent Variable

Table 3 shows the results of the multiple regression analysis with the PHQ-9 score (depressive symptom severity) as the dependent variable, and DOP, age, sex, education years, marital status, past history of psychiatric disease, current psychiatric disease, subjective social status, EPQ-R neuroticism, and CD-RISC (resilience) as the independent variables. Neuroticism, resilience, sex, subjective social status, and past history of psychiatric disease were significantly associated with the depressive symptom severity. Other independent variables, including DOP, were not significantly associated with the PHQ-9 score.

### 3.4. Analysis of the Path Model

The model was analyzed using a path model with the robust maximum likelihood estimation method, as shown in Figure 2, with differences from the optimal PAD (DOP), resilience (CD-RISC), neuroticism (EPQ-R), and depressive symptoms (PHQ-9) as the observed variables.

Regarding the direct effects, DOP showed a negative direct effect on resilience and a positive direct effect on neuroticism (Figure 2). Resilience had a negative direct effect on depressive symptoms, and neuroticism had a positive direct effect on depressive symptoms. However, DOP had no significant direct effect on depressive symptoms. Resilience was negatively associated with neuroticism.

Regarding the indirect effects, DOP had a significant indirect effect on depressive symptoms via reduced resilience (standardized coefficient = 0.020; *p* < 0.05). DOP also had a significant indirect effect on depressive symptoms via increased neuroticism (standardized coefficient = 0.057; *p* < 0.05). That is to say, resilience and neuroticism were mediating variables in the effect of DOP on depressive symptoms. In other words, these results indicated that the greater the difference from the optimal PAD, the more severe the depressive symptoms via the two paths of decreased resilience and increased neuroticism. The coefficient of determination, *R*^2^, was 0.349, i.e., this model explained 34.9% of the variation in depressive symptoms in adult volunteers.

## 4. Discussion

### 4.1. Resilience Mediates the Effect of Regular Physical Activity Habits on Depressive Symptoms

The results of this study showed that there was an optimal PAD (25.7 h/week = ~3.5 h a day) that minimized depressive symptoms, and that excessively long or short PAD was associated with depressive symptoms via resilience and neuroticism. Furthermore, in the model of this study, the direct effect of DOP on depressive symptoms was not significant, indicating that the effects of resilience and neuroticism completely mediated the relationship. To our knowledge, this is the first study to report the mediating roles of resilience and neuroticism in the effect of PAD on depressive symptoms. Zhang et al. reported that resilience partially mediates the effects of physical activity on negative emotional states, namely, depression, anxiety, and stress, in college students [41], which is similar to the finding of this study. Their study differed from ours in that they did not consider depressive symptoms as a specific dependent variable, and their study was limited to a narrow age group of college students. In addition, Yoshikawa et al. reported that exercise habits indirectly affect depressive symptoms through their effects on resilience [26], but their study differed from ours in that they investigated the association between the presence or absence of intense exercise habits and depressive symptoms. In other words, they did not analyze the association between moderate-intensity physical activity or low-intensity physical activity (such as walking) and depressive symptoms. Only 11% of the subjects were regarded as regular exercisers by their standards. The strength of our study is that it differs from previous studies in that we analyzed the habitual physical activity duration (time), which can be determined quickly and subjectively, regardless of the physical activity intensity, and clarified the mediating role of resilience between DOP and depressive symptoms by determining the optimal PAD, which is the time that maximally reduces depressive symptoms.

### 4.2. Neuroticism Mediates the Effect of Regular Physical Activity Habits on Depressive Symptoms

Associations between neuroticism and depression, and between neuroticism and physical activity, have previously been found [42,43]. A long-term population-based study reported that regular moderate- or high-intensity exercise reduces neuroticism and symptoms of anxiety and depression [21]. With regard to the causal relationships, not only resilience but also neuroticism is a factor that causes depression [16]. Exercise has also been shown to alleviate trait anxiety [44], a concept similar to neuroticism. Although this study is cross-sectional, the studies cited above have demonstrated causality, and the validity of the path direction in our study is considered consistent with previous research. The present study also showed similar effects of the optimal PAD, including low-intensity physical activity, on depressive symptoms via neuroticism. Furthermore, a new finding of this study is that not only too-short PAD, but also too-long PAD may be associated with neuroticism and depressive symptoms. Another strength of this study is that we clarified the mediating effect of neuroticism between DOP and depressive symptoms, which has not been described in previous studies.

### 4.3. Interactions of Resilience and Neuroticism

Prior research has suggested that two different mechanisms, namely, neuroticism and resilience, influence vulnerability and protection against depressive symptoms [45]. Previous studies have reported the effects of resilience and neuroticism on depressive symptoms independently, but to our knowledge, there has been no research to date on how these two factors interact to affect depressive symptoms. Recently, it was found that the effects of parenting experienced during childhood on depressive symptoms in adulthood are mediated by both neuroticism and resilience, with resilience acting in an opposite manner to the impact of neuroticism [46]. Our study also found that resilience and neuroticism have opposite mediating effects on depressive symptoms, consistent with this previous study.

### 4.4. Clinical Significance of the Results

The existence of the possible optimal PAD suggests that both too much and too little physical activity can have a worsening impact on depressive symptoms, and therefore, a routine of optimal PAD should be maintained. A similar potential U-shaped association between leisure time physical activity and depressive symptoms has been reported in a previous study [47], which may be an important point to consider when instructing physical activity habits. Given that regular physical activity habits may influence depressive symptoms through resilience and neuroticism, the results of this study suggest that having a regular physical activity habit is important to reduce the risk of developing depressive symptoms. It is also expected that regular physical activity habits will strengthen stress tolerance levels, as they have a positive impact on resilience. It is indicated that resilience is composed of a number of neurobiological and genetic factors, and that interventions focused on reparable factors can increase acquired resilience, and may play an important role as an intervention point for depressive symptoms [48].

Furthermore, as neuroticism is reported to be characterized by negative information processing, strongly associated with depression [49], the improvement of neuroticism through regular physical activity habits is expected to improve negative information processing and alleviate or protect depressive symptoms. In this study, we analyzed the effects of physical activity duration, not physical activity intensity, and included light physical activity, such as walking, which can be easily modified by lifestyle guidance. Therefore, it is expected that interventions for resilience and neuroticism involving a change in physical activity habits will prevent the onset and worsening of depressive symptoms.

## 5. Limitations

Regarding the limitations of this study, first, as our study had a cross-sectional design, causal associations could not be concluded, and hence a long-term prospective study is needed to clarify the causal associations. As an example, doubts about causality could be raised as follows. There is a possibility that a person with a lower depressive level could practice the optimal PAD (~3.5 hours/day) in their daily life. However, it is difficult to imagine that people with a high degree of depression would engage in a lot of exercise despite their depression. Furthermore, in this study, a linear dose–response relationship between depression and physical activity was not confirmed (*p* = 0.985), indicating that at least a decrease in physical activity with depression could not be confirmed. Second, as this study was conducted on adult volunteers, the findings may not be applicable to patients diagnosed with depression. Third, the physical activity duration was assessed by a self-administered questionnaire, which may not reflect an objective physical activity duration. Objective evaluation of physical activity duration will be required to confirm the findings in future studies. Fourth, the use of DOP should be further verified in future studies. The advantage of using a quadratic equation model is that the coordinates and values of the “bottom” can be found. On the other hand, the disadvantage is that it is generally not accurate enough in model fit to approximate biological phenomena with quadratic equations as well as simple linear regression models.

## 6. Conclusions

This study demonstrated that there would be an optimal physical activity duration that reduces depressive symptoms, that both too much and too little physical activity can exacerbate depressive symptoms, and that resilience and neuroticism are mediating factors in the effect of the optimal physical activity duration on depressive symptoms. These findings suggested that physical activity of a variety of intensities for about 3.5 h a day may improve depressive symptoms by enhancing resilience and reducing neuroticism.

## Figures and Tables

**Figure 1 healthcare-11-01900-f001:**
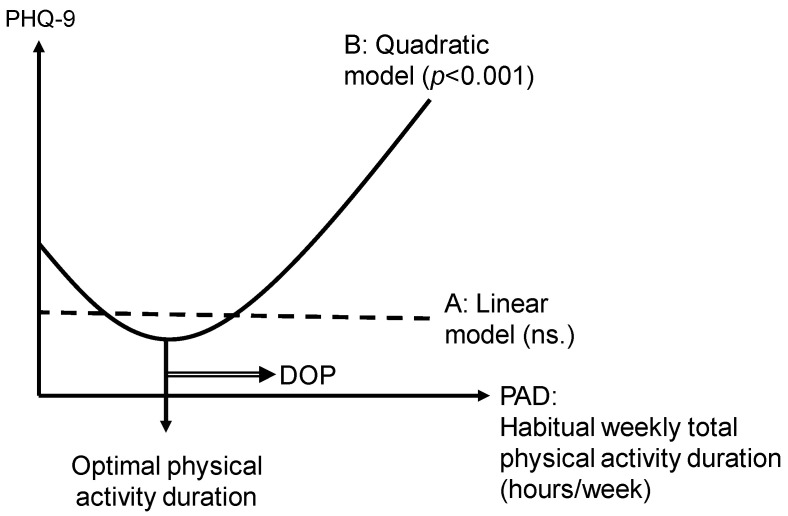
Quadratic equation model regression analysis of the PHQ-9 and the physical activity duration (hours/week). The optimal habitual weekly total physical activity duration (25.7 h/week) indicates a minimum PHQ-9 score among the subjects. DOP, Differences from the optimal habitual weekly total physical activity duration; PHQ-9, Patient Health Questionnaire-9.

**Figure 2 healthcare-11-01900-f002:**
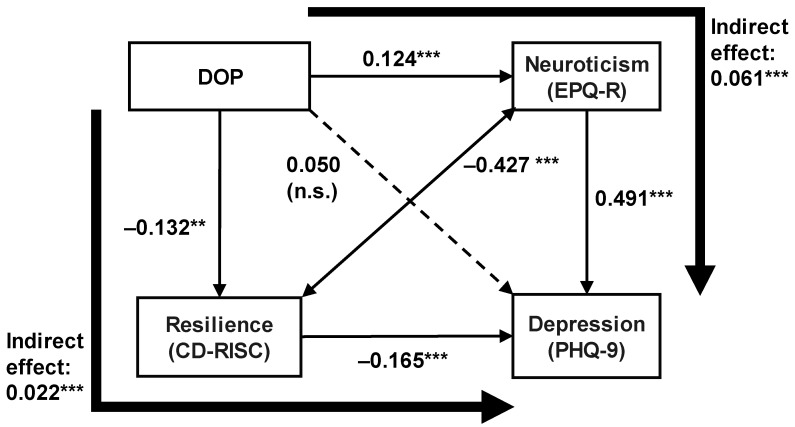
Direct and indirect effects from DOP, neuroticism (EPQ-R), and resilience (CD-RISC) on depression score (PHQ-9) in the path analysis. Numbers show the standardized path coefficients. The thin arrows indicate the direct path effects, and the bold arrows indicate the indirect path effects. ** *p* < 0.01, and *** *p* < 0.001. DOP, Differences from the optimal habitual weekly total physical activity duration; PHQ-9, Patient Health Questionnaire-9; EPQ-R, Eysenck Personality Questionnaire-revised; CD-RISC, Connor-Davidson Resilience Scale.

**Table 1 healthcare-11-01900-t001:** Demographic information and data from each questionnaire and their association with the PHQ-9 score in 526 adult subjects.

Characteristic or Measure	Value (Number or Mean ± SD)	Correlation with PHQ-9 Score (*r*) or Effect on PHQ-9 Score (Mean ± SD of PHQ-9 Score, *t*-Test)
Age (years)	41.2 ± 11.9	*r* = −0.022, *p* = 0.310
Sex (men:women)	228:298	Men 3.3 ± 3.8 vs. women 4.5 ± 4.4, *p* < 0.001 (*t*-test)
Education years	14.8 ± 1.8	*r* = −0.105, *p* = 0.009
Marital status (yes:no)	346:176	Yes 3.5 ± 3.9 vs. no 5.0 ± 4.4, *p* < 0.001 (*t*-test)
Employment (yes:no)	514:9	Yes 4.0 ± 4.2 vs. no 3.3 ± 5.1, *p* = 0.714 (*t*-test)
Subjective social status (1: lowest; 10: highest)	5.2 ± 1.7	*r* = −0.274, *p* < 0.001
Past history of psychiatric disease (yes:no)	62:464	Yes 6.7 ± 5.4 vs. no 3.6 ± 3.8, *p* < 0.001 (*t*-test)
Current psychiatric disease (yes:no)	21:496	Yes 8.1 ± 5.0 vs. no 3.8 ± 4.1, *p* < 0.001 (*t*-test)
Current physical disease (yes:no)	103:423	Yes 4.2 ± 4.7 vs. no 3.9 ± 4.0, *p* = 0.461 (*t*-test)
First-degree relative with psychiatric disease (yes:no)	53:422	Yes 4.5 ± 4.2 vs. no 3.9 ± 4.2, *p* = 0.326 (*t*-test)
EPQ-R-neuroticism score	4.4 ± 3.5	*r* = 0.571, *p* < 0.001
CD-RISC score	55.2 ± 17.4	*r* = −0.386, *p* < 0.001
DOP (hours/week)	19.5 ± 7.5	*r* = 0.124, *p* = 0.004
PHQ-9 score	4.0 ± 4.2	

Data are presented as means ± SD or numbers; *r* = Pearson’s correlation coefficient; EPQ-R, Eysenck Personality Questionnaire-revised; CD-RISC, Connor-Davidson Resilience Scale; DOP, differences from the optimal habitual weekly total physical activity duration; PHQ-9, Patient Health Questionnaire-9.

**Table 2 healthcare-11-01900-t002:** Multiple regression analysis of the effect of physical activity duration on depressive symptoms.

Explanatory Variable for Depression (PHQ-9)	Univariate Model		Quadratic Equation Model		
	Coefficient	*p*-Value	Coefficient	*p*-Value	VIF
Physical activity duration (hours/week)	−0.244 × 10^−3^	0.985	−119.965 × 10^–3^	<0.001	7.575
(Physical activity duration)^2^			2.334 × 10^–3^	<0.001	7.575
*F*-value			6.906	0.001	
Vertex value of habitual physical activity duration (hours/week)			25.70		

PHQ-9, Patient Health Questionnaire-9. The vertex value of physical activity duration was calculated from the significant quadratic equation model.

**Table 3 healthcare-11-01900-t003:** Results of the multiple regression analysis with the PHQ-9 score as the dependent variable, and DOP, age, sex, years of education, marital status, past history of psychiatric disease, current psychiatric disease, subjective social status, EPQ-R, and CD-RISC as the independent variables.

Variable	Beta	*p*-Value	VIF
EPQ-R (neuroticism)	0.466	<0.001	1.398
Past history of psychiatric disease (1: no; 2: yes)	0.086	0.035	1.375
Current psychiatric disease (1: no; 2: yes)	0.079	0.050	1.320
Age	0.079	0.058	1.433
Sex (1: male; 2: female)	0.076	0.040	1.118
Education years	0.023	0.594	1.499
DOP	0.016	0.649	1.067
Marital status (1: unmarried; 2: married)	−0.067	0.080	1.203
CD-RISC score	−0.108	0.010	1.433
Subjective social status	−0.124	0.003	1.394
Adjusted *R*^2^ = 0.390, *F* = 33.266, *p* < 0.001			

Beta = standardized partial regression coefficient; VIF, variance inflation factor; PHQ-9, Patient Health Questionnaire-9; EPQ-R, Eysenck Personality Questionnaire-revised; CD-RISC, Connor-Davidson Resilience Scale.

## Data Availability

The raw data supporting the conclusions of this article will be made available by the authors, without undue reservation.

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
