# Peer review of "Optimal Physical Activity Is Associated with the Reduction of Depressive Symptoms via Neuroticism and Resilience"

_healthcare, 2023, doi:10.3390/healthcare11131900_

Round 1
Reviewer 1 Report
Dear Authors
I congratulate you on your scientifically mature approach to the topic.
I did not find any points in your manuscript that could have been the source of my doubts.
The positive effects of physical activity on mental health have long been confirmed, but the precise timing of this activity seems to be information of major cognitive importance.
Best,
Author Response
Thank you for your kind words and positive feedback on our manuscript.
We deeply appreciate your thoughtful recognition of our work.
Reviewer 2 Report
Dear authors
The article “Optimal Physical Activity Duration is Associated with Depressive Symptoms Through Neuroticism and Resilience” is innovative and brings novel information about the optimal PAD to reduce depressive symptoms.
Some suggestions are provided:
The title of the study must be reformulated. Optimal physical activity is associated with the reduction of depressive symptoms.
Please provide the advantages and disadvantages of the use of the quadratic model.
The abstract is sufficiently informative and presents the main objectives and conclusions of the study. The introduction section is well structured, explaining the variables, the objectives of the study, and the hypothesis. Methods are adequately described. The results were clearly presented. the conclusions supported by the results.
Minor editing of English language required.
Author Response
Dear Reviewer,
Comment #1: The title of the study must be reformulated. Optimal physical activity is associated with the reduction of depressive symptoms.
Response: Thank you for pointing that out. We agree with your idea. In addition, we added "via neuroticism and resilience" as the newness of this paper to the title, considering that the effectiveness of an optimal physical activity duration in reducing depression has already been suggested by some existing studies. We changed the title to "Optimal Physical Activity is Associated with the Reduction of Depressive Symptoms via Neuroticism and Resilience".
Title: revised.
Comment #2: Please provide the advantages and disadvantages of the use of the quadratic model.
Response: Thank you for your comment. As you indicated, we have added the following descriptions about the advantages and disadvantages of the use of quadratic model to the Limitations/Discussion section.
"Fourth, the use of DOP should be further verified in future studies. One advantage of using a quadratic equation model is that the coordinates and values of the "bottom" can be found. On the other hand, the disadvantage is that it is generally not enough in model fit to approximate biological phenomena with quadratic equations as well as simple linear regression models. (page 9, Limitations)"
Comment #3: The abstract is sufficiently informative and presents the main objectives and conclusions of the study. The introduction section is well structured, explaining the variables, the objectives of the study, and the hypothesis. Methods are adequately described. The results were clearly presented. the conclusions supported by the results.
Response: Thank you for your kind words and positive feedback on our manuscript.
For more details, please see the revised manuscript.
Reviewer 3 Report
After carefully reading the manuscript entitled "Optimal Physical Activity Duration is Associated with Depressive Symptoms Through Neuroticism and Resilience", I find that the authors have done an acceptable job. However, there are some comments and recommendations that I would like to make. Before publication can be recommended, improvements to the current iteration are required. Please refer to the comments and recommendations listed below.
The introduction and the statement of the hypothesis is complex to understand, the aim and objective of the study is not clear. It should be more concise.
The methodology is unclear, in particular concerning the sample.
Where were the volunteers obtained from and how were they recruited?
Was a sample size study done?
Were all the tests referred to in the methodology by self-administered responses or how detailed was the data collection process? How many researchers were involved in the collection process?
How long did each interview take? What is the age range of the sample?
If the only exclusion criterion was "severe physical or Psychiatric disease" which was considered as "severe", which number of the sample was excluded.
If the questionnaires are mostly self-administered, wouldn't the level of education be a limitation?
Questionnaires need to be better detailed. Some provide more information than others, such as scores, number of items, etc.
Mean age and SD in the text do not coincide with those expressed in table 1.
Author Response
Dear Reviewer,
Comment #1: The introduction and the statement of the hypothesis is complex to understand, the aim and objective of the study is not clear. It should be more concise.
Response: Thank you for your suggestions. We largely re-organized and shortened the Introduction to make the aim and objective clearer.
Introduction: revised.
Comment #2: The methodology is unclear, in particular concerning the sample. Where were the volunteers obtained from and how were they recruited? Was a sample size study done?
Response: Thank you for your comment. As you indicated, we added the following descriptions to the Subjects and methods section. (Page 3, Methods: 2.1. Subjects)
"Subjects were recruited by convenience sampling through our acquaintances, who were workers and their relatives, at Tokyo Medical University. Additionally, we aimed to collect a sample size of 651 individuals in order to detect an effect size of d=0.2 with 95% significance level."
Comment #3: Were all the tests referred to in the methodology by self-administered responses or how detailed was the data collection process? How many researchers were involved in the collection process? How long did each interview take? What is the age range of the sample?
Response: This study did not involve interviews and instead utilized a paper-based questionnaire survey. Data collection did not involve any interviewers. This study was part of a large-scale questionnaire survey on adult volunteers; many questionnaires were distributed to subjects. It was estimated that it takes 1 hour to complete this survey. The age range was 20 to 77. We have provided additional details about the data collection process in the Subjects and Methods section, specifically on pages 2-3, lines 93-99. (Page 3, Methods: 2.1. Subjects)
"Data were collected by paper-based questionnaire survey without interviews, and questionnaires were returned anonymously by mail. Subjects were recruited by convenience sampling through our acquaintances, who were workers and their relatives, at Tokyo Medical University…mean age: 41.2 ± 11.9 years; age range 20 to 77"
Comment #4: If the only exclusion criterion was "severe physical or Psychiatric disease" which was considered as "severe", which number of the sample was excluded.
Response: Thank you for your question. We did not mention this matter clearly in the original manuscript. We recruited subjects by convenience sampling through our acquaintances, who were workers and their relatives. They were generally healthy and had no severe diseases influencing life functions. Therefore, this exclusion criterion excluded no subjects from recruited subjects. We added the following description to the Methods section. (page 3, Methods, 2.1. Subjects)
“Subjects were generally healthy and had no severe diseases influencing life functions. Therefore, this exclusion criterion excluded no subjects from recruited subjects.”
Comment #5: If the questionnaires are mostly self-administered, wouldn't the level of education be a limitation?
Response: Thank you for your feedback. It is true that subjects who cannot read would not be able to answer the question. The literacy rate in Japan is almost 100%; most subjects' educational levels were high school or more; the questionnaires were simple. We believe that the level of education would not influence the results.
Comment #6: Questionnaires need to be better detailed. Some provide more information than others, such as scores, number of items, etc.
Response: As indicated, we changed the descriptions of questionnaires (EPQ-R, CD-RISC, and PHQ-9) so that the explanations are equalized.
Methods: 2.2. Questionnaires: revised.
Comment #7: Mean age and SD in the text do not coincide with those expressed in table 1.
Response: Thank you for pointing out this matter. We apologize for our mistake. The correct mean age and SD value was 41.2 ± 11.9 as shown in the text. We have corrected the values in Table 1.
For more details, please see the revised manuscript.
Reviewer 4 Report
Personality traits, such as neuroticism, that result in vulnerability to stress, and resilience, a measure of stress coping, are closely associated with the onset of depressive symptoms, whereas regular physical activity habits have been shown to reduce depressive symptoms.
In this study, the authors investigated the mediating effects of neuroticism and resilience between physical activity duration and depressive symptoms.
The effects of these factors on depressive symptoms were analyzed by a covariance structure analysis.
The authors found that the greater the difference from optimal physical activity duration (PAD), the higher the neuroticism and lower the resilience, and more severe the depressive symptoms.
The authors concluded that there was an optimal PAD that reduces depressive symptoms, and that a greater difference from the optimal PAD has increased depressive symptoms through neuroticism and resilience.
This is an interesting study.
I have some minor comments with a pure academic spirit.
1) Insert a clear purpose
2) Avoid short paragraphs (see 2.2.2)
3) 4.1 is not a discussion
4) Check the figures. There is a lot of text. I suggest to move the text into the body of the ms and to smooth some passages
Author Response
Dear Reviewer,
Comment 1): Insert a clear purpose
Response: We have changed the Introduction section including the last paragraph about the purpose and hypothesis of this study in response to the reviewer’s comment. The “purpose” would become clear now, as follows. (page 2, Introduction)
"Therefore, we focused on the mediating effects of neuroticism and resilience, and hypothesized that there is an optimal habitual physical activity duration regarding its effect on depressive symptoms and that the effects of habitual physical activity on depressive symptoms are mediated by its impact on neuroticism and resilience. To test this hypothesis, we conducted a cross-sectional study using a questionnaire survey on adult volunteers, and tested the hypothesis by a path analysis. The purpose of this study was to test the mediating effects of neuroticism and resilience between optimal habitual physical activity and the severity of depressive symptoms in adult volunteers."
Comment 2) Avoid short paragraphs (see 2.2.2)
Response: Thank you for pointing out this matter. We have merged the text of the Questionnaires section and re-organized the descriptions. (page 3, Methods)
“2.2. Questionnaires
Age, sex, education years, currently marital status (yes/no), currently employment status (yes/no), subjective social status (1 to 10) [31], existence of the past history of psy-chiatric or physical disease (yes/no), and first-degree relative with psychiatric disease (yes/no) were asked as demographic variables.
The shortened version of the neuroticism subscale, Eysenck Personality Question-naire-revised (EPQ-R) [32] was used to assess neuroticism. This subscale consists of 12 items, and each item is assessed using a 2-point scale of "yes=1" and "no=0". The total score of the questionnaire was used for the study analyses. Our previous research has es-tablished the reliability and validity of the Japanese version of the EPQ-R [33].
The Connor-Davidson Resilience Scale (CD-RISC) [9] has been widely used in recent years to assess resilience, and the Japanese version of the CD-RISC was used in this study. Its reliability and validity have been confirmed in the previous research [34]. The CD-RISC comprises 25 items, which are rated on a 5-point scale (0 to 4), and the total score was used for analysis in this study.
The Patient Health Questionnaire-9 (PHQ-9) is a self-administered depression rating scale [35] . Depressive symptoms in the previous 2 weeks are assessed. The PHQ-9 con-sists of 9 items, and each item is assessed using a 4-point scale (0 to 3). In this study, the total score of the Japanese version of the PHQ-9 was used for analysis. The translation and validation of the Japanese version of the PHQ-9 for reliability and validity were conducted by Muramatsu et al [36].”
Comment 3) 4.1 is not a discussion
Response: Thank you for pointing out this matter. We have modified a discussion section as follows. (Page 7, Discussion)
“4.1. Resilience mediates the effect of regular physical activity habits on depressive symptoms
The results of this study showed that there is an optimal PAD (25.7 hours /week = ~3.5 hours a day) that minimized depressive symptoms, and that excessively long or short PAD is associated with depressive symptoms via resilience and neuroticism. Furthermore, in the model of this study, the direct effect of DOP on depressive symptoms was not significant, indicating that the effects of resilience and neuroticism completely mediated the relationship. To our knowledge, this is the first study to report the mediating roles of resilience and neuroticism in the effect of PAD on depressive symptoms. Zhang et al. reported that resilience partially mediates the effects of physical activity on negative emotional states, namely, depression, anxiety, and stress in college students [41], which are similar to the finding of this study.”
Comment 4) Check the figures. There is a lot of text. I suggest to move the text into the body of the ms and to smooth some passages
Response: Thank you for pointing out this matter. We have modified the figure. The formula in the figure is now in the manuscript (3.2. Association of PHQ-9 scores with demographic and clinical data, and questionnaire data of the study population).
Figure 1: modified.
For more details, please see the revised manuscript.
Round 2
Reviewer 3 Report
The authors have satisfactorily addressed the reviewers's comments however, it still needs a few more revisions. It is necessary to describe the tests in more detail to facilitate the reader's understanding. Especially, in relation to scoring.
Author Response
Dear Reviewer,
Thank you for your valuable advice. We have added detailed descriptions about the scoring of the measurements in this study.
(p.3 Subjects and methods: 2.2. Questionnaires)
The shortened version of the neuroticism subscale, Eysenck Personality Questionnaire-revised (EPQ-R) [32] was used to assess neuroticism. This subscale consists of 12 items, and each item is assessed using a 2-point scale of "1: yes" and "0: no". The total score (range: 0-12) of the questionnaire was used for the study analyses. Our previous research has established the reliability and validity of the Japanese version of the EPQ-R [33].
The Connor-Davidson Resilience Scale (CD-RISC) [9] has been widely used in recent years to assess resilience, and the Japanese version of the CD-RISC was used in this study. Its reliability and validity have been confirmed in the previous research [34]. The CD-RISC comprises 25 items, which are rated on a 5-point scale (0: not true at all, 1: rarely true, 2: sometimes true, 3: often true, and 4: true nearly all of the time) based on how the subject has felt over the past month, and the total score (range: 0-100) was used for analysis in this study.
The Patient Health Questionnaire-9 (PHQ-9) is a self-administered depression rating scale [35] . Depressive symptoms in the previous 2 weeks are assessed. The PHQ-9 consists of 9 items, and each item is assessed using a 4-point scale (0: not at all, 1: several days, 2: more than half the days, and 3: nearly every day). In this study, the total score (range: 0-27) of the Japanese version of the PHQ-9 was used for analysis. The translation and validation of the Japanese version of the PHQ-9 for reliability and validity were conducted by Muramatsu et al [36].
For more details, please see the revised manuscript.